# Copper Foam as Active Catalysts for the Borylation of *α, β*-Unsaturated Compounds

**DOI:** 10.3390/ijms23158403

**Published:** 2022-07-29

**Authors:** Kewang Zheng, Miao Liu, Zhifei Meng, Zufeng Xiao, Fei Zhong, Wei Wang, Caiqin Qin

**Affiliations:** 1College of Chemistry and Materials Science, Hubei Engineering University, Xiaogan 432000, China; kewang1104@126.com (K.Z.); liumiao0727@163.com (M.L.); meng16301@163.com (Z.M.); chemhbeu@163.com (Z.X.); qincq@hbeu.edu.cn (C.Q.); 2Hubei Key Laboratory of Biological Resources and Environmental Biotechnology, Wuhan University, Wuhan 430079, China

**Keywords:** porous materials, copper foam, catalytic, aqueous phase reaction, B_2(_pin_)2_, *a*, *β*-unsaturated ketone

## Abstract

The use of simple, inexpensive, and efficient methods to construct carbon–boron and carbon–oxygen bonds has been a hot research topic in organic synthesis. We demonstrated that the desired *β*-boronic acid products can be obtained under mild conditions using copper foam as an efficient heterogeneous catalyst. The structure of copper foam before and after the reaction was investigated by polarized light microscopy (PM), scanning electron microscopy (SEM), and transmission electron microscopy (TEM), and the results have shown that the structure of the catalyst copper foam remained unchanged before and after the reaction. The XPS test results showed that the Cu(0) content increased after the reaction, indicating that copper may be involved in the boron addition reaction. The specific optimization conditions were as follows: CH_3_COCH_3_ and H_2_O were used as mixed solvents, 4-methoxychalcone was used as the raw material, 8 mg of catalyst was used and the reaction was carried out at room temperature and under air for 10 h. The yield of the product obtained was up to 92%, and the catalytic efficiency of the catalytic material remained largely unchanged after five cycles of use.

## 1. Introduction

The addition reaction of diborons with olefin is an effective method to obtain organoboron compounds [1,2,3]. Organoboronic acid compounds are important intermediates in drug molecules and organic synthesis, where C-B bonds could be converted into C-C, C-H, and C-O bonds by simple reactions [4,5,6]. Therefore, the efficient construction of carbon–boron bonds, especially the synthesis of chiral organoboronic acid compounds, has been a hot topic of interest for researchers.

Current methods for the synthesis of C-B bonds include C-X bond boronization [7], C-H bond boronization [8], C-O bond boronization [9], and addition boronization of olefins [10,11,12,13]. Of them, addition boronization of olefins has received a lot of attention from organic chemists, due to the easy source of olefin reaction substrates and the variety of reaction types. Earlier studies used simple olefins as substrates and catecholborane as a monofunctional borohydride reagent to construct C-B bonds. The first rhodium-catalyzed olefin addition reaction was reported by Mannig et al. [14] in 1985. Burgess et al. [15,16,17] subsequently investigated rhodium-based catalytic systems for the asymmetric borohydration of olefins and obtained olefin boron addition products by the addition of chiral ligands with an enantioselectivity of up to 76%. Compared to simple olefins, *a, β*-unsaturated compounds are more reactive and could be used as substrates to selectively form C-B bonds at the *β*-position, so methods [18,19,20,21,22,23] to achieve conjugated boron addition reactions via *a, β*-unsaturated compounds have been developed successively. Transition metals such as rhodium [24,25], palladium [26,27], platinum [28,29], nickel [30,31] and copper [32,33] are mostly used as catalysts. The use of copper salts or copper complexes as catalysts will greatly reduce the cost of the reaction and therefore have an important place in the conjugation borylation of *α, β*-unsaturated compounds.

In 2000, Ito et al. [34] has achieved the first conjugated boronization of *α*, *β*-unsaturated ketones in DMF using copper sulphonate as a catalyst and bis(pinacolato)diborane polar solvents (B_2_(pin)_2_) as a boron source. In 2008, Lee [35] achieved the asymmetric synthesis of *α*, *β*-unsaturated esters and nitriles in a methanol/tetrahydrofuran mixture using cuprous chloride as the catalyst and adding sodium tert-butoxide as the base. In 2016, Ding [36] reported the highly chemoselective catalytic reduction of C=C in *a, β*-unsaturated ketones with the CuBr/B_2_pin_2_ system. The boron addition reaction in water is more in line with the green requirements [12]. Copper(II) salt has become a new hot spot for the study of boron addition reactions as the copper(II) salt is more stable in water than the copper(I) salt. The groups of Santos [22], Kobayash [37], and Casar et al. [38] have successively studied the boron addition reactions of *α, β*-unsaturated compounds using various copper(II) salts in water. Among them, the Kobayashi group prepared three ligands that were stable and active in the aqueous phase. In recent years, some Cu(0) nanoparticles have also been used as catalysts [23,39] for boron addition reactions of *α, β*-unsaturated compounds.

Foam metals are a new type of functional material with a large number of pores within their structure. Foam metals have excellent physical properties, such as porosity, high specific surface area, high mechanical strength, and heat resistance, making them a promising candidate for applications such as carriers for catalytic materials [39,40,41,42].

In previous work, we immobilized copper salts or copper nanoparticles on Zeolite [43], chitosan microspheres [44], chitosan films [45], and cellulose [46] and they all had high catalytic activity in the boron addition reaction. As a continuation of the above work, we believe that copper foam can also be used directly as catalysts for boronization processes. Therefore, we report here the performance of copper foam as a highly active and recyclable catalyst for boron addition to *α, β*-unsaturated acceptors in aqueous media. We demonstrate that the desired *β*-boronic acid product can be obtained under mild conditions using copper foam as an efficient heterogeneous catalyst.

## 2. Results and Discussion

### 2.1. Optimization of Reaction Conditions

Many variables are involved in the yield of the boron addition reaction product. In this experiment, the reaction conditions were optimized in the hope of increasing the yield of the boron addition product. Considering the results of our previous work, we chose the specific reaction conditions set as follows: 4-methoxychalcone (**1a**, 0.2 mmol), copper foam (10 mg), solvent (2 mL), and the reaction was carried out for 12 h in air and at room temperature.

It has been reported that proton-type solvents [37,46], such as methanol, can better promote the boron addition reaction of *α, β*-unsaturated receptors. Therefore, we first chose the proton solvent as the solvent for the reaction. The different polar solvents are screened in Table 1, Entry 1–5, and the yields of the polar solvents methanol, ethanol i-PrOH, DMF, and water were 67%, 53%, 25%, <5%, and 8%, respectively. When non-protonic solvents ether, toluene, dichloromethane, and tetrahydrofuran were used as solvents for the reaction, none of the reaction yields exceeded 20% (Table 1, Entry 6–9). This also proves that in this reaction, the proton source can accelerate the reaction. When we chose acetone as the reaction solvent, the yield of the product was high (Table 1, Entry 10). Then, we investigated the effect of methanol, acetone, and water solvent mixture on the reaction. The experimental results showed (Entry 11 and Entry 12) that the addition of water can significantly increase the yield. Considering the solvent benefit of water, we then screened the ratio of acetone to water (Entry 13–15), and it can be seen that the yield of the product was 92% at the optimal ratio of acetone to water of 2:1 (*v/v*). The catalyst dosage has a certain effect on the yield of the reaction. We chose acetone and water (2:1, *v/v*) as the reaction solvent; the reaction did not proceed without copper foam, and the yield could reach 93% when the catalyst dosage was 8 mg (Entry 16–18). When the reaction time was 10 h, the yield could reach 92%, but a longer reaction time that this yielded basically no change. In summary, the optimum reaction conditions were template substrate 4-methoxychalcone (0.2 mmol), 8 mg catalyst, 1.35 mL CH_3_COCH_3_ and 0.65 mL H_2_O at room temperature for 10 h.

### 2.2. Substrate Expansion under Optimal Reaction Conditions

Numerous reports [43,44,45,46] have shown that the oxidation reaction of organoboron compounds is an equivalent transformation. To verify the utility of the copper foam catalytic system, we employed a “one-pot” strategy to extend the substrate by direct conversion of *α, β*-unsaturated acceptors to *β*-hydroxy compounds under optimal reaction conditions. The substrate expansion is shown in Table 2 below.

First, we investigated the catalytic activity of copper foam for monosubstituted chalcone derivatives (**1a**–**1g** and **1i**–**1k**). We found that monosubstituted chalcone derivatives possessing electron-donating groups such as methoxy, methyl, and benzyloxy groups gave high yields of *β*-hydroxy products (80–94%). Monosubstituted substrates possessing electron-absorbing groups on the benzene ring, such as trifluoromethyl, and fluorine have slightly lower reactivity (80–89%). The substitution group in the interposition has no effect on the activity of the catalyst. We then investigated the catalytic activity of copper foam for the disubstituted chalcone derivatives (**1l**–**1p**), and excellent yields (82–98%) were obtained for both boron addition reactions. From the above experiments, it can be seen that different positions and types of groups on the benzene ring have little effect on the yields. High yields (46–92%) were obtained for naphthalene ring substrates (**1q**), aliphatic substrates (**1r**–**1v**), and heterocyclic thiophene substrates (**1x**, **1y**). The slightly lower yields for aliphatic and heterocyclic thiophene substrates could be attributed to the reduced activity of the substrates due to the reduced conjugation of the substrates. When *α, β, γ, δ*-unsaturated ketones substrates (**1w**) were catalyzed with copper foam, we found that 1–4 addition products were generated, which indicates that the copper foam material has good stereoselectivity. In conclusion, the results of the substrate adaptation study showed that the boron addition reactions of the chalcone series of derivatives (**1a**–**1y**) all yielded the target products in excellent yields under optimal reaction conditions. The ^1^H NMR and ^13^C NMR spectra of products **2a**, **3aa**, and **3a-x** are shown in Appendix A.

### 2.3. Gram-Scale Synthesis Reaction of **1a**

We investigated the conversion of 4-methoxychalcone (**1a**) as a substrate to *β*-hydroxy compound (**3a**) by a one-pot method in a scale-up reaction (Figure 1). It was shown that when the reactant (**1a**) was 5 mmol, the corresponding product (**3a**) was still obtained in a high yield of 90%.

### 2.4. Catalyst Reuse

To demonstrate the stability of copper foam, cycling reuse tests were performed under optimal conditions in this study. After each cycling test, the copper foam was collected, washed three times alternately with water and acetone, and dried under vacuum conditions of 60 °C for 12 h. As can be seen in Figure 1, the yield of copper foam remained essentially unchanged after five cycles.

### 2.5. Characterization of Catalytic Materials and Mechanistic Studies

We investigated the state changes in copper foam before and after the reaction by means of polarizing microscope (PM), X-ray diffraction (XRD), scanning electron microscope (SEM), transmission electron microscope (TEM) and X-ray photoelectron spectroscopy (XPS) analytical tests.

#### 2.5.1. Polarizing Microscope Analysis

The copper foam was glued on the slide, and the brightness of the light was gradually adjusted from weak to strong to identify the color and glossy nature of the copper foam specimen, and the measurement condition was 100 or 200 times magnification. The results showed that the copper foam had many macroporous structures, the surface color of copper became darker and the luster decreased after the reaction. This indicates that the copper foam was involved in the boron addition reaction. The observation results of polarizing microscopy are shown in Figure 2.

#### 2.5.2. SEM-EDS Analysis

Figure 3 shows the SEM images of copper foam before and after the reaction at different magnifications. It can be seen from the figure that the pore size of copper foam was approximately 1 mm. Compared with the raw material, the surface of the copper foam skeleton after five reactions was still smooth and the pore structure did not change significantly, indicating that the copper foam has a strong physical stability. The relative contents of copper and oxygen in the copper foam were detected by EDS analysis. Appendix A shows the atomic percentages measured by EDS. Compared with unreacted copper foam, the oxygen content of the reacted copper foam has decreased. The experimental results showed that the ratio of copper to oxygen increased slightly after the reaction, indicating that a small amount of copper oxide may have been turned into copper.

#### 2.5.3. XRD Analysis

To study the changes in the catalyst before and after the reaction, we characterized the unreacted copper foam, the copper foam after the first reaction, the copper foam after the second reaction, and the copper foam after the fifth reaction by XRD diffraction tests. As can be seen in Figure 4, all copper foams show characteristic diffraction peaks at 43.35°, 50.46°, and 74.17° of 2θ, which correspond to the (111), (200), and (220) crystal planes of the copper standard card, respectively (PDF: 04-0836). Additionally, the (200)/(111) ratios of these copper foams were in the range of 0.43–0.52, which is close to the (200)/(111) ratio of the copper standard card (0.46), indicating that the copper foam surfaces were not oxidized. No shedding of copper was observed in the solution throughout the reaction. Although a deepening of the color of the solution was observed during the reaction, the characteristic peaks of Cu^2+^ could not be observed from XRD because of the small amount of Cu^2+^ formed and dissolved in the solution.

#### 2.5.4. XPS Analysis

To investigate the valence change and distribution of copper elements, we characterized the unreacted copper foam, the copper foam after the first reaction, and the copper foam after the fifth reaction by XPS tests. As shown in Figure 5a, two main peaks appear in the spectrum at binding energies of 932.6 eV and 952.54 eV, attributed to the Cu 2p3/2 and Cu 2p1/2 peaks of Cu^0^, respectively [47], and the binding energy peaks of 934.2 eV and 955.1 eV point to the Cu 2p3/2 and Cu 2p1/2 peaks of Cu^2+^, respectively. As shown in Figure 5b,c, the Cu 2p3/2 and Cu 2p1/2 peaks attributed to Cu^0^ are still evident, while the Cu 2p3/2 and Cu 2p1/2 peaks of divalent Cu gradually decrease. This may be the involvement of the copper foam in the boron addition reaction and the oxide on the copper surface entering the solution.

#### 2.5.5. TEM Analysis

The surface structures of copper foam and copper foam after primary reaction were analyzed by transmission electron microscopy. From Figure 6a,b, it can be seen that the crystal structure of the unreacted copper foam is relatively clear with a lattice spacing of 0.22 nm, which should correspond to the (1, 1, 1) crystal plane of face-centered cubic structure copper. As can be seen from Figure 6c,d, the crystal structure of the reacted copper foam is still relatively clear with a lattice spacing of 0.22 nm, which is not significantly different from that of the unreacted copper foam. Combined with the XPS test results, this indicates that the state of the reacted copper is unchanged.

#### 2.5.6. Isotopic Effects of Reactions

To investigate the reaction mechanism of copper foam in catalyzing the boron-addition reaction of *α, β*-unsaturated acceptors, we performed deuteration experiments using 4-methoxychalcone as raw material using the one-pot method, as shown in Table 3. The results are in general agreement with the previous experiments in Zhou’s report [46]. Both MeOH and H_2_O were used as solvent and proton sources during the catalytic cycle reaction. The yield of the boron addition product decreased with the addition of deuterium reagent, while at the *β*-position methylene of the carbonyl group, an H atom was replaced by a D atom. This result suggests that both H and D atoms can participate in the catalytic cycle as proton sources, and the protonation step of the reactants may be the rate-determining step in the overall catalytic process, with the protonation rate of H atoms being greater than that of D atoms.

### 2.6. Mechanistic Studies

Based on the previous and current experimental results of our group, we propose a possible reaction mechanism as shown in Figure 7. In the first step, B_2_pin_2_ breaks the B-B bond catalyzed by copper foam (A) to form the active copper–boron metal complex (B) and the by-product Bpin-OH. In the second step, the active copper–boron metal complex (B) undergoes an addition reaction guided by the carbonyl group in the *α, β*-unsaturated acceptor to give the intermediate C. Then, the intermediate C undergoes a transition state rearrangement to produce intermediate D. Finally, intermediate D exchanges protons in the solvent to produce the target product E while regenerating the catalytic material.

### 2.7. Comparison of the Catalytic Performance of Different Copper Catalytic Materials

So far, several copper catalysts with high catalytic performance have been developed and applied to different coupling reactions of diborons with *α, β*-unsaturated compounds (Table 4). Zhu et al. [48,49] have studied the reaction performance of the catalysts in the presence of chiral ligands L2 or L3 using chitosan@Cu(OH)_2_ or CuO catalysts and found that Cu(OH_)2_ and CuO catalyzed the generation of *β*-addition products almost quantitatively. Zhou et al. [50] have found that the reaction required only 2 h to obtain 95% of the product using carbon black supported Cu^0^ material as a catalyst. Other studies from our group [43,44,45,46] also showed that both lower-order and higher-order copper are excellent materials for the boron addition reaction. Compared with the above catalysts, the copper foam has a good catalytic effect (92% yield), does not require additional support and ligands, and is more easily separated, making it a more suitable catalyst for a large number of reactions.

## 3. Materials and Methods

### 3.1. Materials

Copper foam and all of the above reagents were purchased from Energy Chemical. NMR (Bruker Avance III 400 Hz, Berlin, Germany) was used to verify the structure of the products. Scanning Electron Microscopy (JEOL, JSM-6510, Tokyo, Japan) were used to measure the morphology and sizes of the modified chitosan microspheres. The FTIR spectra of the microspheres were obtained using a Nicolet iS5 Spectrophotometer (Thermo, Austin, TX, USA) to investigate possible interactions between the microspheres and the nano-copper. Microscopy (TEM, JEOL-2100F, Tokyo, Japan), X-ray Photoelectron Spectroscopy (XPS, ESCALAB 250xi, Thermo, USA) and X-ray Diffractometry (XRD, S2, RIGAKU, Tokyo, Japan) were used to obtain the elemental valence of the catalyst copper. Purification of the product was carried out using column chromatography (silica gel, 200–300 mesh).

### 3.2. Arylboronic Acid Self-Coupling Reaction

Copper foam (010 mg), 4-methoxychalcone (**1a,** 0.2 mmol) and bis (Pinacolato) diboron (0.3 mmol) were added to the reaction flask and a solvent mixture of acetone and water (v:v = 1:1) was added to react for 12 h, at room temperature. It was filtered, washed with ethyl acetate and the solvent was evaporated to give a mixture containing *β*-boronic acid compound **2a**. Then, the mixture was added to a mixed solution of THF and H_2_O containing 0.2 mmol of NaBO_3_.4H_2_O and stirred for 4 h, at room temperature. After stopping the reaction, extraction and fractionation, vacuum-drying was performed. The target product **3a** was isolated using a mixture of ethyl acetate and petroleum ether as eluent (v:v = 1:4–1:10). The reaction route is shown in Figure 8. The structure of the product was verified by NMR.

## 4. Conclusions

In this paper, copper foam was used directly as a catalyst for the borylation addition reaction of *α, β*-unsaturated compounds under mild conditions. The experimental results show that copper foam is an easy separation, easy recovery and high efficiency catalyst, and the yield of the product is still up to 90% after five cycles. This catalyst is an important reference for the green synthesis and industrial application of C-O compounds.

## Data Availability

Supporting data can be obtained from the corresponding authors.

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
