# Peer review of "Copper Foam as Active Catalysts for the Borylation of α, β-Unsaturated Compounds"

_ijms, 2022, doi:10.3390/ijms23158403_

Round 1

Reviewer 1 Report

The manuscript "Copper foam as active catalysts for the borylation of α, β-unsaturated compounds" by F. Zhong and W. Wang et al. reports on the catalytic performance of copper foam in borylation of a specific substrate - 4 methoxychalcone in optimized conditions. The effort of the authors to successfully conduct these studies based on previous experience as mentioned in this work is to be appreciated. I have only some minor observations:

- The Abstract should be rewritten by mentioning the characterization analyses used to prove the results;

- the first sentence in Introduction it should be replaced with one reffering to catalytic coupling reaction of boronic acids;

- in Results, the authors should add some results of their previous work as they mentioned in first paragraph of this section;

- in the second paragraph of the Results Section, replace "proton solvents" with "polar solvents: if you mentioned DMF, which is aprotic polar sovent;

- why authors choose the ratio 2:1 acetone:water, if in Table 1 the yield in single solvent is lower?

- the influence of the catalyst amount should be discussed, because as it is shown in Table 1 at lower amount the yield in higher at a short time, why?

-in Figure 7, there is a mechanism of transformation of the substrate including only the influence of the solvents? The authors also mentioned some copper transformation during the reaction, this ones should be included in this figure.

Based on these considerations I recommend the publication of this paper after Minor revision!

Author Response

Point 1: The Abstract should be rewritten by mentioning the characterization analyses used to prove the results;

Response 1: Thank you for your suggestion. We have revised the Abstract.

Abstract: The use of simple, inexpensive, and efficient methods to construct carbon-boron and carbon-oxygen bonds has been a hot research topic in organic synthesis. We demonstrated that the desired β-boronic acid products can be obtained under mild conditions using copper foam as an efficient heterogeneous catalyst. The structure of copper foam before and after the reaction was investigated by polarized light microscopy (PM), and scanning electron microscopy (SEM), transmission electron microscopy (TEM), and the results have shown that the structure of the catalyst copper foam remained unchanged before and after the reaction. The XPS test results showed that the Cu(0) content increased after the reaction, indicating that copper may be involved in the boron addition reaction. The specific optimization conditions were as follows: CH3COCH3 and H2O were used as mixed solvents, 4-methoxychalcone was used as the raw material, 8 mg of catalyst was used and the reaction was carried out at room temperature and under air for 10 h. The yield of the product obtained was up to 92% and the catalytic efficiency of the catalytic material remained largely unchanged after five cycles of use.

Point 2: the first sentence in Introduction it should be replaced with one reffering to catalytic coupling reaction of boronic acids;

Response 2: Thank you for your suggestion. We have already replaced a sentence about the boron addition reaction in the first sentence of the introduction.

The addition reaction of diborons with olefin is an effective method to obtain organoboron compounds[1-3]. Organoboronic acid compounds are important intermediates in drug molecules and organic synthesis, where C-B bonds could be converted into C-C, C-H, and C-O bonds by simple reactions [4-6].

Point 3:  in Results, the authors should add some results of their previous work as they mentioned in first paragraph of this section;

Response 3: Thank you for your suggestion. We have added some results of our previous work in Results.

2.7 Comparison of the catalytic performance of different copper catalytic materials

Table 4 Comparison of catalytic performance of copper foam catalysts with other copper materials

Entry

catalyst

Support

Ligand

Yield(%)

1[48]

Cu(OH)2

-

L1

93%

2[48]

Cu(OH)2

Chitosan

L1

90%

3[48]

Cu(OH)2

Chitosan

L2

100%

4[48]

Cu(OH)2

Chitosan

-

90%

5[49]

CuO

-

L3

96%

6[49]

Cu(OAc)2

Chitosan

-

91%

7[46]

CuI

Cellulosic

-

96%

8[50]

Cu2O

-

-

NR

9[50]

Cu

Carbon black

-

95%

10[44]

Cu

Chitosan

-

93%

11[45]

Cu

Chitosan/PVA

-

87%

12

copper foam

-

-

92%

So far, several copper catalysts with high catalytic performance have been developed and applied to different coupling reactions of diborons with α, β-unsaturated compounds (Table 4). Zhu et al [48, 49] have studied the reaction performance of the catalysts in the presence of chiral ligands L2 or L3 using chitosan@Cu(OH)2 or CuO catalysts and found that Cu(OH)2 and CuO catalyzed the generation of β-addition products almost quantitatively. Zhou et al [50] have found that the reaction required only 2 h to obtain 95% of the product using carbon black supported Cu0 material as a catalyst. Other studies from our group [43-46] also showed that both lower-order and higher-order copper are excellent materials for the boron addition reaction. Compared with the above catalysts, the copper foam has a good catalytic effect (92% yield), does not require additional support and ligands, and is more easily separated, making it a more suitable catalyst for a large number of reactions.

Point 4: in the second paragraph of the Results Section, replace "proton solvents" with "polar solvents: if you mentioned DMF, which is aprotic polar sovent;

Response 4: Thank you for your suggestion. We have corrected it.

The different polar solvents were screened in Table 2, Entry 1-5, the yields of the polar solvents: methanol, ethanol i-PrOH, DMF, and water were 67%, 53%, 25%, <5%, and 8%, respectively.

Point 5:why authors choose the ratio 2:1 acetone:water, if in Table 1 the yield in single solvent is lower?

Response 5: The solubility of the reactants of chalcone in water is very low and one of the effects of adding organic solvents is to increase the solubility of the reactants. In our previous literature [43-46], some other solvent mixtures, such as methanol plus water and tetrahydrofuran plus water, were used and good results were obtained in the experiments. In this paper, acetone and water were also chosen as a solvent mixture to verify that the organic solvent (acetone) increased the solubility of the reactants, while water was used to provide protons to facilitate the reaction.

Point 6: the influence of the catalyst amount should be discussed, because as it is shown in Table 1 at lower amount the yield in higher at a short time, why?

Response 6: Thank you for your suggestion. Because of the oxide on the surface of copper foam, it is impossible to measure the copper content accurately, so we only briefly discussed the amount of copper foam added in the reaction (6mg,8mg, and 10mg). We have tried to clean the oxide on the surface of copper foam with hydrochloric acid, but the strength of the cleaned copper foam decreased significantly and it was easily broken when stirred, which led to a troublesome subsequent separation operation. In the follow-up study, we intend to improve the research method to investigate the effect of catalyst dosage on the reaction without affecting the mechanical strength of the copper foam. If the copper foam was pure copper, the amount of copper foam added to the reaction was 8 mg (0.125 mmol), which corresponds to 62.5 % of the catalyst added to the raw material. Our other studies [43, 45, and 46] showed that 60-82% of the product could be obtained in 4-6 h in the presence of copper catalysis with chalcone as the reaction substrate. The study by Zhou et al [50] also showed that the yield of the product could reach 81% in 2 h of reaction using copper as a catalyst.

Point 7: in Figure 7, there is a mechanism of transformation of the substrate including only the influence of the solvents? The authors also mentioned some copper transformation during the reaction, this ones should be included in this figure.

Response 7: Thank you for your suggestion. According to the reference [1], both low-order and high-order cuprates were used as catalysts to obtain products with regioselectivity. The possible reason for this is the generation of a boron copper species (intermediate 1), which is regioselective for terminal alkynes.

 Intermediate 1

There are some copper transformations during the reaction, but due to the limited experimental conditions, we could not obtain more accurate information, so only the effect of the solvent is discussed in the article.

Reference 1 Takahashi K, Ishiyama T, Miyaura N. Addition and coupling reactions of bis (pinacolato) diboron mediated by CuCl in the presence of potassium acetate. Chemistry Letters, 2000, 29(9): 982-983.

Reviewer 2 Report

This work reported that β-boronic acid products can be obtained under mild conditions using copper foam as an efficient heterogeneous catalyst. The yield of the product obtained was up to 92% and the catalytic efficiency of the catalytic material remained largely unchanged after five cycles of use. Overall, the manuscript is well organized and presents sufficient scientific insights. But there are some issues that must be addressed before publication.

1. Usually the size of micropore is less than 2 nm, thus, the microporous structure cannot be observed by polarizing microscope.

2. The present catalytic performance should be compared with previously published paper and copper powders.

Author Response

Point 1: Usually the size of micropore is less than 2 nm, thus, the microporous structure cannot be observed by polarizing microscope.

Response 1: Thank you for your advice. We have revised it.

“The results showed that the copper foam had many macroporous structures”.

Point 2: The present catalytic performance should be compared with previously published paper and copper powders.

Response 2: Thank you for your suggestion. We have added some results of our previous work in Results.

2.7 Comparison of the catalytic performance of different copper catalytic materials

Table 4 Comparison of catalytic performance of copper foam catalysts with other copper materials

Entry

catalyst

Support

Ligand

Yield(%)

1[48]

Cu(OH)2

-

L1

93%

2[48]

Cu(OH)2

Chitosan

L1

90%

3[48]

Cu(OH)2

Chitosan

L2

100%

4[48]

Cu(OH)2

Chitosan

-

90%

5[49]

CuO

-

L3

96%

6[49]

Cu(OAc)2

Chitosan

-

91%

7[46]

CuI

Cellulosic

-

96%

8[50]

Cu2O

-

-

NR

9[50]

Cu

Carbon black

-

95%

10[44]

Cu

Chitosan

-

93%

11[45]

Cu

Chitosan/PVA

-

87%

12

copper foam

-

-

92%

    So far, several copper catalysts with high catalytic performance have been developed and applied to different coupling reactions of diborons with α, β-unsaturated compounds (Table 4). Zhu et al [48, 49] have studied the reaction performance of the catalysts in the presence of chiral ligands L2 or L3 using chitosan@Cu(OH)2 or CuO catalysts and found that Cu(OH)2 and CuO catalyzed the generation of β-addition products almost quantitatively. Zhou et al [50] have found that the reaction required only 2 h to obtain 95% of the product using carbon black supported Cu0 material as a catalyst. Other studies from our group [43-46] also showed that both lower-order and higher-order copper are excellent materials for the boron addition reaction. Compared with the above catalysts, the copper foam has a good catalytic effect (92% yield), does not require additional support and ligands, and is more easily separated, making it a more suitable catalyst for a large number of reactions.

     We did not choose powdered copper as a catalyst for the reaction mainly because of the difficulty of subsequent operations of powdered copper as a catalyst, including extraction and recovery.
